

# The platelet and coagulation function parameters in late pregnancy are associated with preeclampsia and its severity: a single-center retrospective study

Jindi Zhang, Jie Liu and Pei Wang

Department of Obstetrics, Hangzhou Womenís Hospital, Hangzhou, Zhejiang Province, China

## ABSTRACT

**Objective**. This study aimed to explore the correlation between platelet and coagulation function during late pregnancy with preeclampsia (PE) as well as its severity, and concurrently establish an effective risk model to predict PE.

**Methods**. As a retrospective case-control study, this research encompassed a sample of 594 patients of late pregnancy in Hangzhou Women's Hospital from January 2021 to April 2024, including 198 cases diagnosed with mild-PE, 198 with severe-PE, and 198 healthy pregnant women. Utilizing both univariate and multivariate analysis, this study conducted a comparison of demographic and laboratory data among these groups to uncover the correlation between platelet parameters, coagulation function, and PE as well as the severity. Ultimately, a receiver operating characteristic (ROC) curve was drawn to evaluate the predictive power of the risk model for PE.

**Results**. Univariate analysis revealed statistically significant differences in platelet count (PC), mean platelet volume (MPV), plateletcrit (PCT), thrombin time (TT), and fibrinogen mean (FIB). Multivariate analysis showed that MPV (adjusted odds ratio (AOR) = 1.882 for mild-PE group and AOR = 2.141 for severe-PE group) and TT (AOR = 3.071 for mild-PE group and AOR = 4.154 for severe-PE group) were associated with PE. The area under curve (AUC) of the risk model of MPV combined with TT, as depicted by the ROC curve, amounted to 0.827, exhibiting a sensitivity of 0.657 and a specificity of 0.879.

**Conclusions**. MPV and TT are independent risk factors for PE and are associated with its severity. The risk model consisting of MPV and TT possesses a certain predictive capability for PE.

# INTRODUCTION

As one of the prevalent pregnancy complications with a global incidence rate ranging from 5% to 8%, preeclampsia (PE) exhibits the potential to continuously escalate in severity as the gestational period advances (*Xing et al., 2024*). A number of patients who

Corresponding author
Pei Wang, peiwang85@163.com

experiences severe symptoms are diagnosed with severe PE. This group of patients face a significantly heightened risk of poor prognosis for both the mother and the child (*Shenhav et al., 2024*). As a result, these cases are highly prioritized in clinical practice, necessitating close attention and management. Unfortunately, there is no definitive diagnostic tool or therapeutic approach for PE at present (*Hasbini et al., 2024*).

The primary clinical management strategy for PE evolves around pharmacological interventions, monitoring of blood pressure whilst a battery of routine laboratory evaluations (*Bijl et al., 2022*). Recent studies have pinpointed several biomarkers that can either stand alone or be combined to forecast PE, including placental growth factor (PLGF), soluble Fms like tyrosine kinase 1 (sFlt-1), sFlt-1/PLGF ratio, as well as placental protein-13 (PP13), pregnancy associated plasma protein-A (PAPP-A), *etc* (*Ormesher et al., 2020*; *Jeon et al., 2021*). While these biomarkers exhibit remarkable sensitivity or specificity, their clinical application is hindered by their low economic viability (*Park et al., 2020*).

During normal pregnancy, the blood of pregnant women attains a hypercoagulable state, playing a crucial role in mitigating postpartum hemorrhage (*Lidan et al., 2019*). However patients with PE experience abnormal activation of the coagulation system, abnormal consumption of platelets and coagulation factors, sometimes leading to impaired coagulation function even the coexistence of thrombosis and bleeding (*Liu et al., 2020*). Current research results indicated that increased platelet consumption serves as an early characteristic in patients with PE, suggesting a certain correlation between the reduction of platelets and the occurrence of PE. *Zeng & Liao, (2022)* observed that the thrombin time (TT) in PE patients was higher than that of healthy pregnant women. *Limonta, Intra & Brambilla (2021)* discovered that PE patients exhibit elevated levels of D-dimer, which possessed a certain predictive value for PE. Therefore, despite their status as routine hematological assessments, platelet and coagulation function tests potentially encompass a wealth of clinical information, which perhaps can be leveraged clinically for early identification of PE, anticipating the progression, performing effective therapeutic interventions and ultimately improving the prognosis of maternal and child.

This study aimed to explore the correlation between platelet parameters, coagulation function in late pregnancy and PE, further establish a risk model and evaluate its predictive value for PE in the light of receiver operating characteristic (ROC) curves.

## MATERIALS AND METHODS

### Participants

A total of 396 patients with PE were selected from 32,128 patients in the late stages of pregnancy (≥28 weeks) in Hangzhou Women's Hospital (Hangzhou Maternal and Child Health Care Hospital) from January 2021 to April 2024, including 198 cases diagnosed with mild-PE, 198 with severe-PE. Meanwhile, 198 healthy parturients from the same period were included and incorporated into the control group (Fig. 1). All demographic and clinical data of the participants were sourced from the hospital information system (HIS). Due to the retrospective design and the absence of identifiable information in the specimens and laboratory data utilized, the ethics committee exempted this study from

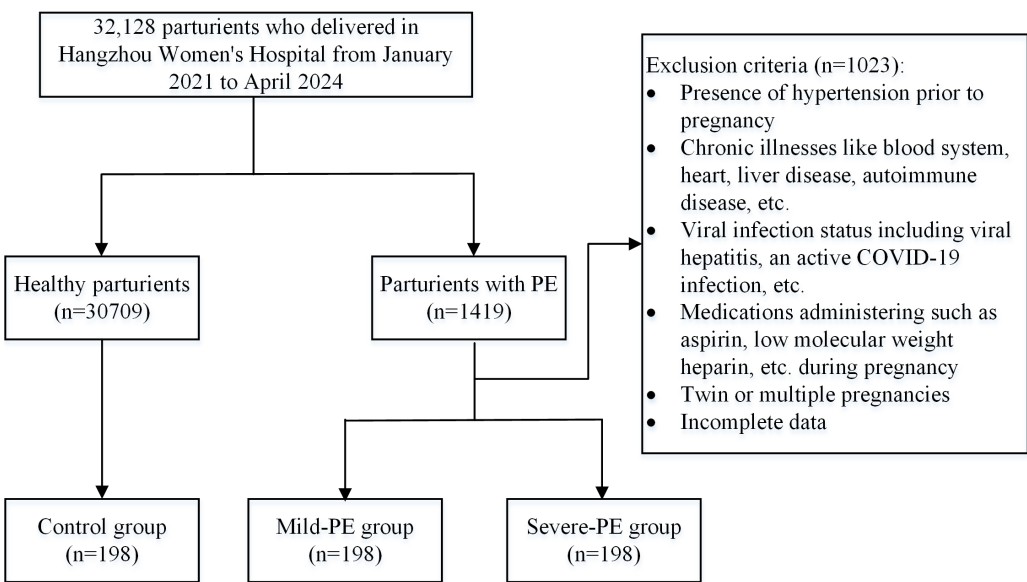

**Figure 1** **Flow chart of the study population.** This is the process of including research subjects.

the informed consent requirement. This study was approved by the Ethics Committee of Hangzhou Women's Hospital (Medical Ethics Review A No. 2024-127). In addition, the study was conducted in accordance with the Declaration of Helsinki.

## Diagnostic and exclusion criteria
### Diagnostic criteria for PE (*American College of Obstetricians and Gynecologists, 2020*)

A systolic blood pressure $\geq$ 140 mmHg and/or a diastolic blood pressure $\geq$ 90 mmHg after the 20th week of pregnancy, accompanied by any one of the followings:

(1)  urinary protein levels $\geq$ 0.3 g/24 h, or an urine protein-to-creatinine ratio $\geq$ 0.3.

(2)  A random urine protein test result $\geq$ (++) (when quantitative urine protein testing is not feasible).

(3)  Proteinuria is absent yet there is involvement of vital organs such as the heart, lungs, liver, kidneys, abnormal changes in the blood system, digestive system, nervous system or placental fetal involvement.

### Diagnostic criteria for severe-PE

Severe-PE is defined as PE with serious manifestations. Serious manifestations include any one of the following situations:

(1) a systolic blood pressure $\geq$ 160 mmHg and/or a diastolic blood pressure $\geq$ 110 mmHg;

(2) decreased platelet count ($<100 \times 10^9$/L);

(3) liver dysfunction (the serum transaminase levels are more than double the normal range) or severe persistent upper abdominal pain that can't be explained by other diseases;

(4) renal dysfunction (blood creatinine level greater than 1.1 mg/dl or creatinine concentration stands more than double the normal range in the absence of other kidney diseases);

(5) pulmonary edema;

(6) newly emerging headaches (which can't be relieved by medication and explained by other diseases);

(7) visual impairment;

*Exclusion criteria:*

(1) presence of hypertension prior to pregnancy;

(2) chronic illnesses like blood system, heart, liver disease, autoimmune disease, *etc.*;

(3) viral infection status including viral hepatitis, an active COVID-19 infection, *etc.*;

(4) medications administering such as aspirin, low molecular weight heparin, *etc.* during pregnancy;

(5) twin or multiple pregnancies;

(6) incomplete data.

## Clinical data collection

The demographic characteristics of all cases were collected, including gestational age, body mass index (BMI), gravidity, parity, baseline blood pressure (tested within 12th weeks of pregnancy, including systolic blood pressure (SBP) and diastolic blood pressure (DBP)), and fasting blood glucose (FBG).

The blood samples of each participant were collected in a resting and fasting state during hospitalization in late pregnancy (≧28 weeks). Data on platelet parameters (including platelet count (PC), mean platelet volume (MPV), plateletcrit (PCT) and platelet distribution width (PDW)) and coagulation function indicators (including activated partial thromboplastin time (APTT), thrombin time (TT), fibrinogen (FIB), prothrombin time (PT) and D-dimer) were gathered.

## Instruments and reagents

All laboratory data of platelet parameters and coagulation function was tested by the automated five-part differential blood cell analyzer (Sysmex XN-2000, Japan) and fully automatic coagulation analyzer (ACL-TOP, Instrumentation Laboratory, USA), fibrinogen and D-dimer reagent, blood clotting tetrachoric single kit. The detection procedure was carried out according to the instructions.

## Quality control

All testing personnel and auditors have completed rigorous pre-employment training, secured qualification certificates issued by health institutions, and regularly engage in professional external quality assessment initiatives. Furthermore, the standard requirements and testing methodologies for both internal and external quality control within the laboratory have been implemented in line with our previous research reports (*Zhang, Wang & Liu, 2025*).

## Statistical analysis

Spss27.0 (IBM Corp., Armonk, NY, USA) was used for the statistical analysis of data. The one-sample Kolmogorov–Smirnov test was applied to evaluate the normality of data in each group, which with skewed distribution were described as median and percentile [M(P$_{25}$-P$_{75}$)]. Univariate analysis was performed by Kruskal–Wasllis test on quantitative data and analysis of variance (ANOVA) on qualitative data. $P < 0.05$ was considered statistically significant. Multivariate analysis was conducted *via* multiple regression analysis, as ordered regression analysis was deemed unsuitable due to the failure of the parallelism test. All the demographic data (excluding FBG) and parameters related to platelet and coagulation function (encompassing PC, MPV, PCT, PDW, APTT, TT, FIB, PT and D-dimer) were incorporated into the regression equation. The adjusted odds ratio (AOR) obtained could reflect the strength of the correlation between relevant indicators and PE. Those that were statistically significant within each group were selected to generate ROC curves. The cutoff value and area under the curve (AUC) were determined from ROC curves to evaluate the predictive value for PE. And the optimal cutoff point along with its corresponding sensitivity, specificity and Youden index were calculated. When $P < 0.05$, the risk model with the maximum AUC as well as higher sensitivity and specificity had better predictive value.

# RESULTS

## Demographic comparisons

There was no significant statistical difference in terms of FBG among these groups ($Z = 3.23$, $P > 0.05$). However, statistically significant differences were observed on maternal age ($Z = 12.18$, $P = 0.002$), BMI ($Z = 66.49$, $P < 0.001$), gravidity ($F = 14.11$, $p < 0.001$)), parity ($F = 122.84$, $P < 0.001$), SBP ($Z = 147.88$, $P < 0.001$) and DBP ($Z = 162.48$, $P < 0.001$, Table 1).

## Comparison of platelet and coagulation function parameters

The result of the Kolmogorov–Smirnov test presented no statistically significant difference in either PDW ($Z = 1.52$. $P = 0.468$), APTT ($Z = 2.96$, $P = 0.228$), PT($Z = 4.89$, $P = 0.087$) or D-dimer ($Z = 4.65$. $P = 0.098$) across these groups (Table 2). There were statistically significant differences in PC ($Z = 28.59$), MPV ($Z = 34.35$), PCT ($Z = 22.25$), TT ($Z = 176.61$) and FIB ($Z = 23.42$) among the three groups (all the $P < 0.001$), indicating that the levels of corresponding indicators were not completely the same among the groups (Table 2).

## Results of multiple regression analysis

The outcomes of multiple logistic regression analysis revealed that MPV (AOR = 1.882 (1.118–3.167), $P = 0.017$) and TT (AOR = 3.071 (2.221–4.248), $P < 0.001$) exhibited significant associations with mild-PE, whereas MPV (AOR = 2.141 (1.250–3.666), $P = 0.006$) and TT (AOR = 4.154 (2.962–5.826), $P < 0.001$) were also factors associated with severe-PE in comparison to the control group (Table 3).

**Table 1  Univariate analysis of demographic data among these groups.**

| Indicators | Control group (n = 198) | Mild-PE group (n = 198) | Severe-PE group (n = 198) | Z/F | P |
|---|---|---|---|---|---|
| Maternal age (years) | 30.00 (28.00–33.00) | 29.00 (27.00–32.00) | 31.00 (28.00–33.00) | 12.18 | 0.002[1] |
| BMI (Kg/m$^2$) | 25.54 (23.87–27.52) | 27.99 (25.89–30.62) | 27.61 (25.56–30.48) | 66.49 | <0.001[2] |
| Gravidity | | | | 14.11 | <0.001[2] |
| 1 | 108 (54.55) | 136 (68.69) | 105 (53.03) | | |
| ≧2 | 90 (45.44) | 62 (31.31) | 93 (46.97) | | |
| Parity | | | | 22.84 | <0.001[2] |
| 0 | 134 (67.68) | 174 (87.88) | 148 (74.75) | | |
| ≧1 | 64 (32.32) | 24 (12.12) | 50 (25.25) | | |
| SBP (mmHg) | 109.50 (101.00–118.25) | 124.00 (116.00–129.00) | 123.00 (118.00–129.00) | 147.88 | <0.001[2] |
| DBP (mmHg) | 63.00 (58.00–70.25) | 75.00 (69.00–79.00) | 73.00 (69.00–78.00) | 162.48 | <0.001[2] |
| FBG (mmol/L) | 4.11 (3.93–4.28) | 4.11 (3.87–4.47) | 4.18 (3.87–4.44) | 3.23 | 0.199 |

Notes.
The quantitative data were expressed by median and percentile [M (P$_{25}$-P$_{75}$)], and qualitative data were expressed by n (%).
[1] $P < 0.05$.
[2] $P < 0.001$.
PE, preeclampsia; BMI, body mass index; SBP, systolic blood pressure; DBP, diastolic blood pressure; FBG, fasting Blood Glucose.

**Table 2  The univariate analysis of platelet and coagulation function parameters.**

| Indicators | Control group (n = 198) | Mild-PE group (n = 198) | Severe-PE group (n = 198) | Z | P |
|---|---|---|---|---|---|
| PC (×10$^9$/L) | 192.00 (167.00–227.50) | 194.00 (159.00–239.25) | 170.00 (139.50–205.50) | 28.59 | <0.001 |
| MPV (fl) | 10.10 (9.40–11.03) | 10.60 (9.80–11.60) | 10.90 (10.00–12.20) | 34.35 | <0.001 |
| PCT (%) | 19.55 (17.38–22.10) | 20.90 (17.58–24.23) | 18.75 (15.60–22.20) | 22.25 | <0.001 |
| PDW (%) | 16.60 (16.30–16.80) | 16.50 (16.20–16.80) | 16.50 (16.30–16.80) | 1.52 | 0.468 |
| APTT (s) | 26.45 (25.48–27.80) | 26.65 (25.20–28.10) | 26.90 (25.60–28.50) | 2.96 | 0.228 |
| TT (s) | 15.60 (15.20–16.20) | 16.70 (15.90–17.50) | 17.60 (16.55–18.35) | 176.61 | <0.001 |
| FIB (g/L) | 4.36 (3.86–4.79) | 4.49 (3.92–4.96) | 4.03 (3.37–4.69) | 23.42 | <0.001 |
| PT (s) | 10.65 (10.20–11.00) | 10.40 (10.10–10.98) | 10.50 (10.10–10.90) | 4.89 | 0.087 |
| D-dimer (ug/L) | 2080.00 (1455.00–2942.50) | 1795.00 (1282.50–2427.50) | 1940.00 (1160.00–3025.00) | 4.65 | 0.098 |

Notes.
The quantitative data were expressed by median and percentile [M (P$_{25}$-P$_{75}$)].
PC, platelet count; MPV, mean platelet volume; PCT, plateletcrit; PDW, platelet distribution width; APTT, activated partial thromboplastin time; TT, thrombin time; FIB, fibrinogen; PT, prothrombin time.

## Predictive value of MPV, TT and their combination for PE

As displayed by the ROC curves and based on the MPV and TT, the AUCs were 0.641 (95% confidence interval (CI) [0.594–0.688]) and 0.810 (95% CI [0.774–0.845]), respectively (Fig. 2 and Table 4). The aforementioned multiple logistic regression analysis elucidated that MPV and TT served as independent risk factors for PE. Additionally, the ROC curve about the combination of MPV and TT possessed the largest AUC expressed as 0.827 (95% CI [0.793–0.862]), along with a sensitivity of 65.7% and a specificity of 87.9% (Fig. 2, Table 4).

**Table 3  Multiple regression analysis of platelet and coagulation function parameters.**

| Group | Indicators | B | SE | Wald | df | P | AOR (95% CI) |
|---|---|---|---|---|---|---|---|
| Mild-PE group | PC | 0.009 | 0.012 | 0.506 | 1 | 0.477 | 1.009 (0.985–1.034) |
| | MPV | 0.632 | 0.266 | 5.669 | 1 | 0.017[1] | 1.882 (1.118–3.167) |
| | PCT | −0.059 | 0.118 | 0.254 | 1 | 0.614 | 0.942 (0.748–1.187) |
| | PDW | −0.975 | 0.533 | 3.351 | 1 | 0.067 | 0.337 (0.133–1.071) |
| | APTT | 0.020 | 0.072 | 0.078 | 1 | 0.780 | 1.020 (0.886–1.175) |
| | TT | 1.122 | 0.165 | 45.984 | 1 | <0.001[2] | 3.071 (2.221–4.248) |
| | FIB | 0.198 | 0.181 | 1.198 | 1 | 0.274 | 1.219 (0.855–1.738) |
| | PT | −0.161 | 0.155 | 1.088 | 1 | 0.297 | 0.851 (0.628–1.152) |
| | D-dimmer | 0.000 | 0.000 | 1.199 | 1 | 0.273 | 1.000 |
| Severe-PE Group | PC | −0.001 | 0.013 | 0.002 | 1 | 0.966 | 0.999 (0.974–1.025) |
| | MPV | 0.761 | 0.274 | 7.691 | 1 | 0.006[1] | 2.141 (1.250–3.666) |
| | PCT | −0.082 | 0.123 | 0.449 | 1 | 0.503 | 0.921 (0.724–1.172) |
| | PDW | −1.974 | 0.576 | 11.729 | 1 | 0.001[1] | 0.139 (0.045–0.430) |
| | APTT | 0.054 | 0.077 | 0.504 | 1 | 0.478 | 1.056 (0.908–1.228) |
| | TT | 1.424 | 0.173 | 68.071 | 1 | <0.001[2] | 4.154 (2.962–5.826) |
| | FIB | −0.134 | 0.199 | 0.453 | 1 | 0.501 | 0.875 (0.592–1.292) |
| | PT | −0.290 | 0.205 | 2.006 | 1 | 0.157 | 0.748 (0.501–1.118) |
| | D-dimmer | 0.000 | 0.000 | 2.248 | 1 | 0.134 | 1.000 |

**Notes.**

Setting Control Group as the reference

[1] $P < 0.05$

[2] $P < 0.001$

PC, platelet count; MPV, mean platelet volume; PCT, plateletcrit; PDW, platelet distribution width; APTT, activated partial thromboplastin time; TT, thrombin time; FIB, fibrinogen; PT, prothrombin time.

# DISCUSSION

Preeclampsia-eclampsia represents a severe manifestation of gestational hypertension, with PE being the initial phase of this condition (*Lucena et al., 2019*). A growing body of research underscores the potential for PE, regardless of its severity, to result in significant adverse outcomes (*Tjahyadi et al., 2023*). Consequently, despite the clinical practice of categorizing PE into mild and severe forms for diagnostic and management convenience, it is imperative that both are afforded equal weight and attention in clinical decision (*Rahman, Anwar & Mose, 2024*). At present, the diagnosis of PE primarily revolves around a combination of medical history, clinical symptom, and auxiliary examinations (*Wainstock & Sheiner, 2022*). Nevertheless, there exists an absence of a highly sensitive, accurate, and cost-effective auxiliary diagnostic tool in clinical practice for early identification and personalized therapy. The main findings of this study were that patients with PE exhibit a significantly elevated BMI compared to their healthy pregnant counterparts, the level of PDW inversely correlates with PE severity whereas MPV and TT exhibit a positive correlation. Furthermore, risk prediction models that integrate MPV and TT demonstrate a promising capacity to predict PE.

A fundamental pathological alteration in PE involves oxidative stress and immune factor-mediated endothelial cell damage, which fosters vasoconstriction and spasms

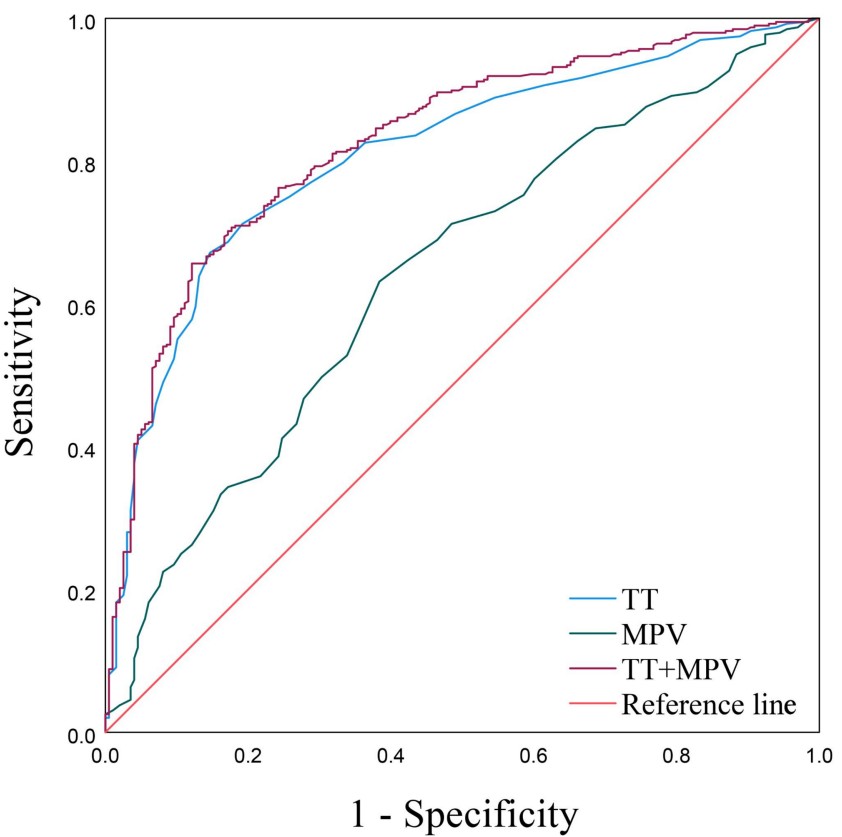

**Figure 2** ROC curves of MPV, TT and TT+MPV.

**Table 4** The value of MPV, TT and the combination for predicting PE.

| Indicators | Youden | Sensitivity | Specificity | Cut-off | AUC | 95% CI | P |
|---|---|---|---|---|---|---|---|
| MPV | 0.247 | 0.631 | 0.616 | 10.450 | 0.641 | 0.594–0.688 | <0.001 |
| TT | 0.525 | 0.672 | 0.854 | 16.550 | 0.810 | 0.774–0.845 | <0.001 |
| MPV+TT | 0.535 | 0.657 | 0.879 | 0.732 | 0.827 | 0.793–0.862 | <0.001 |

Notes.
MPV, mean platelet volume; PDW, platelet distribution width; TT, thrombin time; AUC, area under the curve; CI, confidence interval.

(*Wright & Nicolaides, 2019*). This damage to vascular endothelial cells exposes collagen fibers within the vessel wall, subsequently triggering platelet aggregation, adhesion, and abnormal activation (*Ipek et al., 2021*). This study revealed that the platelet count in severe-PE patients was significantly lower compared to both healthy pregnant women and mild-PE patients. Additionally, the MPV of healthy pregnant women was notably lower than that observed in both PE groups, aligning with previous research findings (*Woldeamanuel et al., 2023*).

The present study revealed no statistically significant difference in PDW between healthy pregnant women and PE patients on the univariate analysis. However, the multivariate analysis showed that PDW emerged as a protective factor against sever-PE. For every
1% increment in PDW, the incidence of sever-PE diminished by 86.1%. This observation contradicts the study findings reported by *Udeh et al. (2024)* who's research discovered that PDW of early-onset preeclampsia patients is higher than that of healthy pregnant women. Furthermore, there exist studies that have failed to establish a significant correlation between PDW and PE, suggesting the need for more comprehensive clinical studies with larger sample sizes to demonstrate the potential associations (*Choudhary et al., 2023*).

The coagulation dysfunction observed in PE patients may intimately correlate with endothelial damage, which subsequently triggers a cascade of pathological alterations (*Combs et al., 2021*). Vascular endothelial injury prompts an abnormal platelet activation, fostering localized ischemia and hypoxia, particularly in vital areas like the placenta, which precipitates placental villus degeneration and necrosis, prompting the release of clotting factors and initiating the exogenous coagulation pathway (*Feldstein et al., 2019*). Additionally, the inadequacy of vascular remodeling results in compromised placental perfusion and the further liberation of inflammatory mediators are reasons for the activation of the coagulation system as well (*Han et al., 2019*). This imbalance in coagulation dynamics can give rise to multiple blood clots during pregnancy, expediting the depletion of prothrombin, FIB, and other coagulation factors (*Narkhede & Karnad, 2021*).

APTT serves as a diagnostic tool to assess the functional activity of the endogenous coagulation pathway (*Deshpande et al., 2024*). PT is employed to quantify the activity of the exogenous coagulation system, specifically measuring the response to exogenous thromboplastin and thus reflecting the extrinsic coagulation pathway's efficacy (*Lefkou et al., 2020*). TT is utilized to comprehensively evaluate the level and functional integrity of fibrinogen, a crucial coagulation factor that plays a pivotal role in the final stages of blood clot formation (*Xu et al., 2021*). Furthermore, D-dimer, a degradation product of cross-linked fibrin, serves as a biomarker for the activity of the fibrinolytic system within the body (*Shao et al., 2021*). Its circulating levels mirror the extent of fibrinolysis, the physiological process responsible for degrading fibrin clots, thereby providing insights into the dynamic balance between coagulation and fibrinolysis (*Hovine et al., 2023*). In this study, the severe-PE group was revealed holding longer TT level than mild-PE group, which in turn longer than the control group (17.60 *vs* 16.70 *vs* 15.60 s, $Z = 176.61$, $P < 0.001$, Table 1). This finding reveals that individuals diagnosed with mild-PE exhibit a prolonged TT compared to healthy pregnant controls, while those with severe-PE display an even shorter TT. The regression analysis results showed that there was a larger AOR of TT for severe PE compared to mild PE. This observation uncovered a positive correlation between TT and the severity of PE, suggesting that TT serves as an independent risk factor of this disease. Moreover, our findings showed a significant decrease in FIB levels among patients with severe-PE, as compared to mild-PE patients and healthy pregnant women. This phenomenon may be attributed to the substantial consumption of FIB during the advanced stages of PE (*Lidan et al., 2019*). The study conducted by Jin Pei Pei et al. revealed that D-dimer levels were found to be significantly lower in patients with PE compared to those of healthy pregnant women (*Jin et al., 2023*). A recent systematic review indicated that patients with PE possessed longer TT and APTT in comparison to healthy pregnant

women, whereas the increase in TT level of PE patients was considered as no statistical significance (*Alemayehu et al., 2024*). In the context of this study, no statistically significant differences were observed in the levels of activated partial thromboplastin time (APTT), prothrombin time (PT), or D-dimer between healthy pregnant women and those diagnosed with PE.

As the multiple logistic regression analysis in this study showed that MPV and TT standed as independent risk factors for PE. The ROC curves uncovered the predictive value of both MPV and TT for PE, with TT exhibiting particularly larger AUC and higher specificity. In addition, the combined risk model incorporating MPV and TT yielded an even larger AUC than that of each indicator along, as well as enhanced sensitivity and specificity.

This study discovered a striking augmentation in BMI among patients with PE as compared to their healthy pregnant counterparts. This observation harmonizes with the previous conclusions drawn by *Mao et al. (2024)* research endeavors. However, studies conducted by *Gong et al. (2022)* revealed that both too high and too low BMI levels can confer an elevated risk of developing PE. Furthermore, it was noteworthy that despite the baseline blood pressure of all three groups falling within the normal range, the systolic and diastolic blood pressure levels of the healthy pregnant women were observed to be lower in comparison to those afflicted with PE. A plausible rationale for these phenomenons was that pregnant women who possess a high BMI were inherently more susceptible to experiencing disruptions in lipid metabolism and abnormal vascular resistance (*Canto-Cetina et al., 2017*). These conditions, in turn, could potentially precipitate elevated blood lipid levels and fluctuations in blood pressure among patients (*Moradi, 2023*). Consequently, these alterations may serve as underlying pathological and physiological mechanisms that contribute to the subsequent development of PE.

There were also several limitations to this study: (I) prior research had implicated smoking and alcohol consumption as potential factors influencing coagulation function, yet this study encountered a significant limitation in assessing their effects on PE due to the exceedingly low proportion of participants reporting alcohol consumption or smoking habits, coupled with the absence of relational PE cases (*Vallée et al., 2025*). Consequently, definitive conclusions regarding the impact of these factors on PE couldn't be drawn from this study. (II) This study didn't encompass cases of PE that manifest prematurely or escalate into severe conditions necessitating pregnancy termination prior to 28 weeks gestation. (III) Patients who failed to provide fasting blood samples owing to rapid disease progression or pregnancy termination resulting from obstetric emergencies during late gestation were excluded from the study. (IV) Lastly, the study's retrospective design and single-center nature, along with the relatively small sample size, restrict the robustness of the findings. Future multi-center studies incorporating larger sample sizes will be necessary to validate these findings.

## CONCLUSION

The levels of MPV and TT in patients with PE are markedly elevated compared to those observed in healthy pregnant women. They emerge as independent risk factors for PE,

demonstrating a positive correlation with the severity of the condition and possessing predictive value for its occurrence and development. Furthermore, the risk prediction models that incorporate MPV and TT exhibits heightened predictive accuracy for PE. These models could potentially pave new avenues for early detection, timely intervention, and personalized therapeutic strategies of PE patients in the future.

## ACKNOWLEDGEMENTS

The author would like to acknowledge the assistance of Baidu ERNIE Bot, an AI-powered natural language processing tool, for improving the language and readability of this manuscript. It was used solely for language refinement, and all content and scientific integrity remain the responsibility of the authors.

### Funding

The authors received no funding for this work.

### Competing Interests

The authors declare there are no competing interests.

### Author Contributions

- Jindi Zhang performed the experiments, prepared figures and/or tables, authored or reviewed drafts of the article, and approved the final draft.
- Jie Liu analyzed the data, prepared figures and/or tables, and approved the final draft.
- Pei Wang conceived and designed the experiments, performed the experiments, authored or reviewed drafts of the article, and approved the final draft.

### Human Ethics

The following information was supplied relating to ethical approvals (i.e., approving body and any reference numbers):

This study was approved by the Ethics Committee of Hangzhou Women's Hospital (Medical Ethics Review A No. 2024-127).

### Data Availability

Raw data is available in the Supplemental Files.

### Supplemental Information

Supplemental information for this article can be found online at http://dx.doi.org/10.7717/peerj.19916#supplemental-information.

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
