# Peer review of "The platelet and coagulation function parameters in late pregnancy are associated with preeclampsia and its severity: a single-center retrospective study"

_PeerJ, doi:10.7717/peerj.19916_

## Round 0.1 · original submission · Major Revisions

**Language Note:** The review process has identified that the English language must be improved. PeerJ can provide language editing services - please contact us at [email protected] for pricing (be sure to provide your manuscript number and title). Alternatively, you should make your own arrangements to improve the language quality and provide details in your response letter. – PeerJ Staff

Reviewer 1 ·

Basic reporting

- The language used in the article is clear and understandable.
- The references used are up-to-date and sufficient.
- Raw data has been shared.

Experimental design

- According to the material-method section, blood samples were taken in the control group when they were hospitalized for delivery and in the PE groups when they were diagnosed. Did this affect the results of the study?
- Pregnancy-related outcomes were not reported.
- How the control group was selected was not reported.
- Instead of mild and severe PE, I think it is correct to classify PE according to whether it has severe features or not.

Validity of the findings

- They did not specify which groups' statistical significance was found.
- How did PDW, which was not statistically significant when the groups were compared, gain significance in multiple regression analysis?

Reviewer 2 ·

Basic reporting

-

Experimental design

-

Validity of the findings

-

Additional comments

Thank you for inviting me to review this manuscript. My specific concerns regarding the paper are included below:
- Studies investigating platelet parameters (like MPV, PC) and coagulation markers (like TT, fibrinogen) in relation to preeclampsia are already abundant in the literature. The association of MPV and TT with preeclampsia has been previously reported multiple times. This study neither presents a new biomarker, a novel method of analysis, nor a substantially innovative risk model.
- The study does not contribute any new pathophysiological insight into the mechanisms of PE or the clinical management of patients.
- Lots of grammatical errors should be corrected. The language of the manuscript should be improved by a native English speaker or a professional editing service.
- The objective is not clear. Is it correlation analysis, risk model development, or both? Kindly state the objective clearly and concisely.
- ‘.. including 198 cases diagnosed with mild-PE, severe-PE, respectively…’ This description is confusing.
- 'Fibrinogen mean' is not a proper termination.
- Sensitivity (0.657) is relatively poor for a clinical prediction model.
- The conclusions overstate the findings. The authors wrote that this model has a certain predictive capability. However, this is not supported by the low sensitivity.

---

## Round 0.2 · accepted · Accept

Thank you for addressing the reviewers' comments in full. Your manuscript is now ready for publication.

Reviewer 1 ·

Basic reporting

- The article uses generally clear and academic English.
- Terminology (MPV, TT, AUC, etc.) is consistent and standardized.
- Pathophysiology of preeclampsia (endothelial damage, hypercoagulability) is well summarized.
- Hypothesis is supported.

Experimental design

- The research topic is appropriate for the aims and scope of the journal.
- Patient selection criteria, laboratory protocols, and statistical analysis (SPSS, ROC curves) are clearly defined.

Validity of the findings

Although the predictive value of the MPV/TT combination has been previously examined in the literature, this study has limited specificity in that it demonstrates the graded association of this combination with severe PE.

Additional comments

The corrections made by the authors are sufficient for me.